# Effects of *Kadsura coccinea* L. Fruit Extract on Growth Performance, Meat Quality, Immunity, Antioxidant, Intestinal Morphology and Flora of White-Feathered Broilers

**DOI:** 10.3390/ani13010093

**Published:** 2022-12-26

**Authors:** Tianlu Zhang, Dong Zhou, Xin Wang, Tian Xiao, Lingxi Wu, Qi Tang, Ying Lu

**Affiliations:** 1College of Horticulture, Hunan Agricultural University, Changsha 410128, China; 2Agriculture and Rural Affairs Bureau of Tongdao County, Huaihua 418500, China; 3National Research Center of Engineering Technology for Utilization Ingredients from Botanicals, Changsha 410128, China

**Keywords:** *Kadsura coccinea* fruit extract, immunity, meat quality, intestinal flora, antioxidant

## Abstract

**Simple Summary:**

*Kadsura coccinea* was planted in large quantities in south China in recent years, and its functions and effects need to be further studied and discovered. We had previously found that *Kadsura coccinea* fruit extract has anti-diarrhea effects on mice, so we want to explore its influence on broiler breeding. The result showed that added 200 mg/kg *Kadsura coccinea* fruit extract to basic feed can replace traditional antibiotics as a growth promoter. It can improve the growth, slaughter performance, and antioxidant and regulate intestinal microorganisms of broilers, which indicated that the *Kadsura coccinea* fruit extract is feasible as a feed additive.

**Abstract:**

This study aimed to determine whether adding *Kadsura coccinea* fruit extract to the diet of broilers could replace antibiotics. For this study, 300 one-day-old AA white feathered broilers were divided into five groups (no sex separated), with six repetitions per group (n = 10), as follows: blank control group (basal feed, CK group), positive drug (basal feed + 300 mg/kg aureomycin, PD group), and *Kadsura coccinea* low-dose, medium-dose, and high-dose groups (basal feed + 100 mg/kg, 200 mg/kg, and 300 mg/kg of *Kadsura coccinea* fruit extract, LD group, MD group and HD group). The experiment period was divided into early (1–21 days) and late (22–42 days) stage. We found that supplementation with *Kadsura coccinea* fruit extract in the diet significantly improved the growth performance of broilers (*p <* 0.05), reduced the feed to meat ratio (*p <* 0.05), reduced the fat percentage (*p <* 0.05), while had no significant effect on meat quality (*p* > 0.05) and *Kadsura coccinea* fruit extract could promote the development of immune organs to different extents, enhance antioxidant capacity, the contents of SOD and GSH-Px in serum were significantly increased (*p <* 0.05), improve the ratio of villus height to crypt depth. Finally, *Kadsura coccinea* fruit extract increased the relative abundance of probiotics and beneficial bacteria (*Bacteroidales*, *NK4A214*, *Subdoligranulum and Eubacterium hallii*) (*p <* 0.05) and reduced the relative abundance of harmful bacteria (*Erysipelatoclostridium*) (*p <* 0.05) in the gut of broilers. Compared with positive drug group, most of the indexes in the medium-dose group were better or had similar effects. We believe that *Kadsura coccinea* fruit extract can be used as a potential natural antibiotic substitute in livestock and poultry breeding programs.

## 1. Introduction

Antibiotics are a class of secondary metabolites produced by microorganisms (including bacteria, fungi, and actinomycetes), animals, and plants, which have anti-pathogenic effects and interfere with the development of other living cells [1]. Since their discovery in the 1920s, antibiotics have played a key role in animal improving growth, feed conversion efficiency, production, and preventing infections [2]. Use of in-feed antibiotics (IFAs) is a common and effective practice in animal husbandry, it can increase productivity and efficiency [3]. With the mass use of IFA, scientists predict that the use of antimicrobials in livestock farming will reach 105,596 ± 3605 tons by 2030 [4]. However, with the widespread growth of animal husbandry, there are concerns that the use of IFAs could lead to the development of antimicrobial resistance, thereby posing a potential threat to human health [5,6]. In recent years, owing to the abuse of antibiotics and drug residues, animal production and food safety issues have been widely considered in all aspects of society. Many countries have legislated the restriction or even prohibition of the addition of antibiotics to feed; the (US Food and Drug Administration (FDA) Silver Spring, MD, USA) set new rules to limit antibiotic use in farm animals in January 2017 [7] and China had stopped the production of commercial feed containing feed additives containing growth promoting drugs (except for traditional Chinese medicine) since July 1, 2020 [8]. Recently, more and more studies have shown that plant chemicals, as effective antibiotic substitutes, can stimulate innate immune cells, reduce oxidative stress, maintain gut integrity, promote the growth of beneficial bacteria, and reduce the negative consequences of intestinal infection, in order to promote poultry, pigs, beef, and dairy products in the production of animal growth performance [6]. The studies on the use various herbs as a natural antibiotics in animal husbandry are being conducted around the world. Therefore, the development of Chinese herbal medicinal resources that can replace antibiotics as feed additives in breeding industry is a potent research topic.

*Kadsura coccinea* (Lem.) A. C. Smith (*K. coccinea*) is a Schisandra species belonging to the Magnoliaceae family, and is mainly distributed in southern China, Thailand, and South Korea. In China, the root of this plant can be used to treat stomach disease, rheumatic bone pain, drop, and bruise pain [9]. The main chemical components of *K. coccinea* are lignans, triterpenes, and flavonoid polyphenols, with a variety of biological activities, including antioxidant, anti-inflammatory, anti-tumor, and anti-HIV effects [10,11,12]. In our previous study, we found that the *K. coccinea* fruit extract exhibited anti-diarrhea effects in mice [13]. In the present study, we investigated the effects of *K. coccinea* fruit extract on growth performance, meat quality, slaughter performance, immune function, cytokines, antioxidant capacity, intestinal morphology, and intestinal microbiota of white-feathered broilers, to provide scientific basis for the use of *K. coccinea* fruit extract t in breeding industry.

## 2. Materials and Methods

### 2.1. Preparation of Kadsura coccinea Fruit Extract

*K. coccinea* fruit was collected in November 2020 at the planting base in Dong Autonomous County of Tongdao, Huaihua City, Hunan Province, China, it was identified by Tang Qi, associate professor of the Department of traditional Chinese medicine resources and development of Hunan Agricultural University. The methods of *K. coccinea* fruit extract selected are as follows: harvested samples were sliced and broken after drying in an oven at 50 °C, the dried fruits (2500 g) were ground to a fine powder and extracted with 10 L 60% ethanol at 50 °C for three times 1 h each, the extracts were combined and concentrated on a rotary evaporator (R1001, Zituo Instrument Equipment Co., Ltd., Zhengzhou, China) under reduced pressure at 55 °C until there was no ethanol completely. Then, the extracting solution was added into 1000 mL AB-8 macroporous adsorption resin column at a flow speed of 2 BV/h and left to stand for 30 min, the resin was washed by water (approximately 8 L) and then eluted with 70% ethanol at a flow speed of 3 BV/h until the eluent is almost colorless. The 70% ethanol eluent was collected and concentrated by rotary evaporation at 55 °C and freeze dried, 41.5 g extract powder was yielded. The above procedure was repeated three times to obtain raw materials for animal experiments.

### 2.2. Determination of the Main Components of K. coccinea Fruit Extracts

The contents of general flavones (Sodium nitrite method) [14], triterpenes (Vanillin-acetic acid method) [15], lignans (Chromotropic acid-concentrated sulfuric acid method) [16] and polysaccharide (Anthrone-concentrated sulfuric acid method) [17] in the extracts of *K. coccinea* fruit were determined using UV spectrophotometry (Spectrophotometer, UV-1900PC, Shanghai Aoyi Instrument Co., Ltd., Shanghai, China).

### 2.3. Animals and Experimental Details

Three hundred 1-day-old healthy AA white feathered broilers (purchased from Hunan Shuncheng Industrial Co., Ltd. (Hunan, China)) with similar body weights (approximately 46g per broiler) were randomly divided into five groups (no sex separated), with six replicates per group and 10 broilers per replicate. The five groups were: blank control group (basal feed, CK Group (Chesterfield, UK)), positive drug (basal feed + 300 mg/kg aureomycin, PD group), and *K. coccinea* low-dose, medium-dose, and high-dose groups (basal feed + 100 mg/kg, 200 mg/kg, and 300 mg/kg of *K. coccinea* fruit extract, LD group, MD group and HD group). The experiment lasted for 42 days, divided into early (1–21 days) and late (22–42 days) growth stage.

The basal diet was formulated according to the “Chicken Feeding Standard” (NY/T33-2004, China). The nutritional requirements were based on the amino acid requirements of NRC1994 broilers. The basal diet composition and nutritional levels are shown in Table 1.

The experiment was carried out in Kaihui Experimental base, Changsha County, College of Animal Science and Technology, Hunan Agricultural University. The coop was thoroughly disinfected and cleaned 1 week before the experiment. Adopts full Enclosed chicken coop, automatic facilities control the temperature, humidity and ventilation inside the coop, the initial temperature is 32 °C, each Weekly temperature drop 2–3 °C, until 20 °C, daily light 23 h, stop for 1 h (21:00–22:00), 1–7 days using plastic cylinder feeding powder feed and drinking water, feed 4 times a day, water 2 times a day; feed granular feed and drinking water with feed tank and water line from 8 to 42 days, change feed once a day. Conduct chicken house hygiene according to routine management procedures.

### 2.4. Details Data and Samples Collection

#### 2.4.1. Growth Performance

The final weight (FW), average daily weight gain (ADG), average daily feed intake (ADFI), ratio of feed to weight gain (F/G) for entire experimental period was calculated.

#### 2.4.2. Serum Index, Immune Organs and Meat Samples

At days 21 and 42, fasted for 12 h, six broilers in each group were randomly selected for jugular vein blood collection using vacuum vessels. After coagulation at room temperature, centrifuge at 3500 RPM for 15 min. The serum samples were separated and stored at −20 °C for further use. After the broilers were killed by bleeding through the carotid, collected the thymus, spleen, liver, bursae and muscle (left pectoral and leg muscles).

#### 2.4.3. Jejunal and Cecum Chyme Samples

Approximately 1 to 2 cm segment of jejunum was removed and separated, rinsed with 0.9% normal saline and fixed in 4% paraformaldehyde for intestinal tissue morphological analysis. Then, cecal chyme samples were collected in 2 mL cryogenic vials and rapidly frozen in liquid nitrogen. All samples were stored at −80 °C until further analyses [18].

### 2.5. Collection Index Determination Method

#### 2.5.1. Slaughter Performance

On the morning of day 42, the broilers were sacrificed by bleeding through the carotid artery after the fasting weight was determined, and hot water depilation. Semi-eviscerated percentage, fully eviscerated percentage, pectoral muscle, leg muscle, and abdominal fat were successively weighed, and the slaughter performance indices were calculated. The specific methods were based on the “Terminology and Measurement and Statistical Methods of Poultry Performance” (NY/T 823-2004, China) (1) Slaughter rate was the ratio of body weight after exsanguination, removal of feathers, foot cuticle, toe shell and beak shell to body weight before death. (2) Semi-evisceration ratio is the ratio of the weight of trachea, esophagus, cyst, intestine, spleen, pancreas, biliary and reproductive organs, stomach contents and keratinous membrane removed on the basis of slaughter ratio to the body weight before death. (3) The fully eviscerated percentage is the ratio of the weight of heart, liver, stomach, lung, abdominal fat, head and feet on the basis of the half-evisceration rate to body weight before death. (4) The pectoral muscle rate is the ratio of the weight of pectoral muscle to the body weight before death. (5) The leg muscle rate is the ratio of the weight of leg muscle to the body weight before death. (6) The abdominal fat percentage rate is the ratio of the weight of abdominal fat to the body weight before death).

#### 2.5.2. Determination of Meat Quality

The specific methods were based on the “Agricultural industry standard of the People’s Republic of China” (NY/T 1180-2004, NY/T 2260-2014, and NY/T 2793-2015, China). L*, a* and b* were measured 45 min after death by colorimeter (Colorimeter, NR20XE; Shenzhen Threenh Technology Co., Ltd., Shenzhen, China). The pH value (45 min and 24 h postmortem) was measured with a pH meter (Testo 205, Teto Instruments International Trading (Shanghai) Co., Ltd., Shanghai, China), and calculated 24 h PH change value. The water loss rate using a filter paper press method and the drip loss was scored based on suspension method, with the leg muscle and chest muscle samples (2.0 cm × 1.5 cm × 1.5 cm) that were weighed, suspended in a plastic bag, held for 24 h, and thereafter reweighed. The shear instrument (Digital display muscle tenderness meter, C-LM4, College of Engineering, Northeast Agricultural University (Harbin, China)) was used to cut the meat along the direction of the muscle fiber and the shear force value was recorded.

#### 2.5.3. Antioxidant Indices and Immune Function

The serum of glutathione peroxidase (GSH-Px), the activities of superoxide dismutase (SOD), total antioxidant capacity (T-AOC), malondialdehyde (MDA), immunoglobulin A (IgA), immunoglobulin G (IgG), immunoglobulin M (IgM), Complement 3 (C3), and Complement 4 (C4) on days 21 and 42 were determined by the commercial kits and an enzyme-linked immunosorbent assay (ELISA) kits (Nanjing Jiancheng Bioengineering Institute, Nanjing, China). A multifunctional enzyme marker (Spark, Hunan Zhike Instrument Equipment Co., Ltd., China) and an incubator (SPX-150BIII, Huanghua Faithful Instrument Co., Ltd., Cangzhou, China) were used in the detection process. For detailed detection methods, please refer to the website: http://www.njjcbio.com/ (accessed on 30 November 2022).

#### 2.5.4. Function Histomorphometry of Jejunum

The jejunum was removed from 4% paraformaldehyde fixing solution, samples were embedded in paraffin, then paraffin sections were performed, and stained with hematoxylin-eosin (HE staining). Selecting 5 intestine sections with complete morphology and clear vision, the microscope image processing software (Case Viewer) was used to measure villus height (VH), crypt depth (CD) and ratio of villus height to crypt depth (V/C).

#### 2.5.5. Serum Mucosa Cytokines Expression

The RNA extracted from the serum was extracted with the RNA extract (Wuhan Xavier Biotechnology Co., Ltd., Wuhan, China), and the extracted RNA was transcribed into CRNA with the RevertAid First Strand cDNA Synthesis kit (Thermo Fisher Scientific (Waltham, MA, USA)), Then, RT-qPCR was performed to obtain the cycle threshold (Ct) values of IL-2 and IFN-γ genes of serum cytokines, and the relative expression levels of IL-2 and IFN-γ genes were calculated by 2^−ΔΔCT^ method.

#### 2.5.6. Analysis of Cecum Microbe

Total DNA was extracted from cecal microbial communities using a DNA kit (Omega Bio-Tek, Hercules, CA, USA). The V3-V4 variable region of the 16S rRNA gene was amplified using the primers 338F (5′-ACTCCTACGGGAG-GCAGCAG-3′) and 806R (5′-GGACTACHVGGGT-WTCTAAT-3′). Sequencing was performed using the Illumina MiSeq PE300 platform (Shanghai Major Biomedical Technology Co., Ltd., Shanghai, China). The nucleotide sequence data reported in this paper have been submitted to NCBI (National Center for Biotechnology Information (Bethesda, MD, USA) nucleotide sequence database and have been assigned the SRA accession number SRR21620835-SRR21620893.

### 2.6. Statistical Analysis

Excel 2010 and SPSS 2.0 were used to process and analyze the data. One-way Analysis of Variance (One-way ANOVA) and Tukey’s Multiple Comparison Method were used to test differences between groups. Significance was set at *p* < 0.05. Data are expressed as mean ± standard deviation.

## 3. Results

### 3.1. Main Chemical Composition Content

The content results of main chemical components of *K. coccinea* fruit extract were shown in Table 2. In the *K. coccinea* fruit extract, the contents of general flavones, triterpene, lignans, polysaccharide were 9.77%, 5.20%, 28.07%, 6.70%, respectively.

### 3.2. Growth Performance

In the early stage (1–21 days), compared with the CK group, the final weight of the PD and LD group were lower (*p* < 0.039 and *p* < 0.015), and ADG of the PD and LD group were lower (*p* < 0.021 and *p* < 0.012), the F/G of the LD group was increased (*p* < 0.002); compared with the PD group, ADFI in the MD group was increased (*p* < 0.030), and F/G in the LD group increased (*p* < 0.049) (Table 3).

In the late stage (22–42 days), compared with CK group, the final body weight and ADG of the other groups increased, while the MD group was increased (*p* < 0.039), and the final weight of the MD group increased by 10.6 percent. The ADG of the PD and MD group were increased (*p* < 0.036 and *p* < 0.008). F/G was highest in the HD group and lowest in the LD group, but with no difference among groups (*p >* 0.05). Compared with the PD group, there was no difference in each index (*p >* 0.05) (Table 3).

With respect to the entire experimental period (1–42 days), compared with the CK group, the ADG of the MD group increased (*p* < 0.039). Compared with the PD group, there was no difference in each index (*p >* 0.05) (Table 3).

### 3.3. Slaughter Performance

Compared with the CK group, the dressing percentage and semi-eviscerated percentage of the four groups increased, and the semi-eviscerated percentage of the MD group increased (*p* < 0.025). The eviscerated percentage and thigh muscle percentage rate of the MD, and HD group were increased but not difference (*p >* 0.05). The abdominal fat percentage decreased in the PD, LD, MD, and HD groups (*p* < 0.001, *p* < 0.003, *p* < 0.002, *p* < 0.001), and the HD group declined by 65%. The breast muscle percentage in the LD and MD groups was decreased (*p* < 0.049, *p* < 0.044). Compared with the PD group, the thigh muscle percentage in the MD and HD groups was increased (*p* < 0.009, *p* < 0.027), no difference was found in other indices (*p >* 0.05) (Figure 1).

### 3.4. Meat Quality

Compared with the CK group, no differences in L*, a*, b*, drip loss, and pressure loss of the leg muscles were found in the four groups (*p >* 0.05). The shear force of the PD and MD group increased (*p* < 0.028 and *p* < 0.038). Compared with the PD group, no difference among the groups was found for the indices (*p >* 0.05) (Figure 2).

Compared with the CK group, the L*, a*, shear force, drip loss, and pressure loss of the four groups showed no difference (*p >* 0.05). The b* and pH 24 h reduction values of the four groups were increased, and the b* values of the PD group were difference (*p* < 0.036); the pH 24 h reduction values of the MD group were different (*p* < 0.05). Compared to the PD group, no difference was found for all indices (*p >* 0.05) (Figure 3).

### 3.5. Immune Organs

In early stage (1–21 days), no differences in the indices of the liver, thymus, and bursa of Fabricius were found between the four groups and the CK group (*p >* 0.05). There spleen index was increased in the HD group (*p* < 0.049). Compared with the PD group, no differences were found in the liver, thymus, and spleen indices among the sample dose groups (*p >* 0.05). The bursa of Fabricius index of the MD group was increased (*p* < 0.032) (Figure 4).

In the late stage (22–42 days), no differences in the indices of the liver, spleen, and bursa were found between the four groups and the CK group. The thymus index of the MD and HD groups was increased (*p* < 0.010 and *p* < 0.012). Compared with the PD group, no differences were found in the indices of the liver, spleen, liver, and bursa of Fabricius with each sample dose group (*p >* 0.05) (Figure 4).

### 3.6. Serum Immunoglobulin

In the early stage (1–21 days), compared with the CK group, the IgG indices of the PD, LD, and HD groups decreased (*p* < 0.001). The IgM index of the LD and HD groups increased (*p* < 0.001). The IgA index in the PD group increased (*p* < 0.003), and that in the LD and HD groups decreased (*p* < 0.001). The C3 indices in the MD and HD groups decreased (*p* < 0.001 and *p* < 0.015), whereas the C4 indices of the HD group increased (*p* < 0.001). Compared with the PD group, the IgG index in the MD group increased (*p* < 0.001), and the IgM indices in the LD and HD groups increased (*p* < 0.001). The IgA indices of the LD and MD groups decreased (*p* < 0.001). The C3 index in the LD group increased (*p* < 0.003), and the C3 index in the MD group decreased (*p* < 0.001). The C4 index of PD group was higher than that of the LD group (*p* < 0.002) but lower than that of the HD group (*p* < 0.004) (Figure 5).

In the late stage (22–42 days), compared with the CK group, the IgG and IgM indices of the four groups were lower (*p* < 0.001). The IgA indices in the LD, MD, and HD groups decreased (*p* < 0.001), and the C3 index of the HD group increased (*p* < 0.001), but the C3 indices of the LD and MD groups decreased (*p* < 0.001 and *p* < 0.022). The C4 indices of the LD and HD groups were decreased (*p* < 0.001 and *p* < 0.007). Compared to the PD group, the IgG index of the MD group was decreased (*p* < 0.001). The IgM indices of the LD, MD, and HD groups increased (*p* < 0.001), and the IgA index decreased (*p* < 0.001). The C3 index was lower than that of the HD group (*p* < 0.001) but higher than that of the LD and MD groups (*p* < 0.001). The C4 indices of the LD and HD groups were lower (*p* < 0.001 and *p* < 0.017) (Figure 5).

### 3.7. Cytokine Expression in Serum

In the early stage (1–21 days), compared with the CK group, the mRNA expression level of IL-2 decreased in the LD, MD, and HD groups (*p* < 0.009, *p* < 0.024, and *p* < 0.009). The mRNA expression level of IFN-γ decreased in the LD groups (*p* < 0.041). Compared to the PD group, the mRNA expression levels of IL-2 in the LD, MD, and HD groups decreased (*p* < 0.003, *p* < 0.009, and *p* < 0.003), and the levels of IFN-γ in the LD, MD, and HD groups decreased (*p* < 0.005, *p* < 0.011, and *p* < 0.007) (Figure 6).

In the late stage (22–42 days), compared with the CK group, the mRNA expression levels of IL-2 and IFN-γ in the MD group were increased (*p* < 0.013 and *p* < 0.011). Compared to the PD group, the mRNA expression levels of IL-2 and IFN-γ in the MD group increased (*p* < 0.031 and *p* < 0.034) (Figure 6).

### 3.8. Serum Antioxidant

In the early stage (1–21 days), compared with the CK group, the SOD content in the PD and MD groups increased (*p* < 0.009 and *p* < 0.003). The MDA content of the HD group was decreased (*p* < 0.029). The GSH-Px content in the MD group increased (*p* < 0.037). Compared with the PD group, no difference was found in the T-AOC and MDA content among the groups. The levels of SOD and GSH-Px in the LD group were lower (*p* < 0.007 and *p* < 0.018) (Figure 7).

In the late stage (22–42 days), compared with the CK group, the T-AOC content in the PD group decreased (*p* < 0.002). MDA content in the HD group was higher (*p* < 0.005). The GSH-Px content in the PD, LD, MD, and HD groups increased (*p* < 0.001, *p* < 0.001, *p* < 0.006, and *p* < 0.001). Compared to the PD group, the T-AOC content in the LD, MD, and HD groups increased (*p* < 0.002, *p* < 0.004, and *p* < 0.012). The MDA content in the HD group increased (*p* < 0.001). The GSH-Px content in the MD group was decreased (*p* < 0.015) (Figure 7).

### 3.9. Intestinal Morphology

In the early stage (1–21 days), compared with the CK group, the villus height of the four groups increased, but no significant difference was found (*p >* 0.05). The V/C of the MD group was increased (*p* < 0.006). Compared to the PD group, the crypt depth of the MD group was lower (*p* < 0.031). The V/C of the MD group was increased (*p* < 0.005) (Figure 8).

In the late stage (22–42 days), compared with the CK group, the villus height and the V/C in the MD and HD groups were increased (*p* < 0.005 and *p* < 0.009). Compared with the PD group, the villus height of the HD groups was increased (*p* < 0.029). The V/C of MD group was increased (*p* < 0.031) (Figure 8).

### 3.10. Intestinal Flora

Figure 9 shows the alpha diversity index of intestinal flora in each group. In the early stage (1–21 days), no significant difference was found in the alpha diversity index of the intestinal microbiota in the four groups compared with the CK group. Compared with the PD group, the alpha diversity of the intestinal microbiota in the LD group significantly decreased (*p* < 0.05), and that of the MD group showed a significant decrease (*p* < 0.01) (Figure 9A). In the late stage (22–42 days), compared with the CK group, the alpha diversity index of intestinal microbiota in the HD group significantly increased (*p* < 0.05), and that in the PD group, the LD and MD groups showed a significant increase (*p* < 0.01). Compared to the PD group, no significant difference was found in LD, MD, HD groups (Figure 9B).

Figure 10 shows the composition of each group at the family level. In the early stage (1–21days), the flora in each group mainly comprised *Ruminococcaceae*, *Lachnospiraceae*, *Clostridia_UCG-014*, *Oscillospiraceae*, and *Erysipelatoclostridiaceae* (Figure 10A). In the late stage (22–42 days), the flora in each group mainly comprised *Lachnospiraceae*, *Ruminococcaceae*, *Barnesiellaceae*, *Bacteroidaceae*, and *Oscillospiraceae* (Figure 10B).

Figure 11 shows the flora heat map of each group at the genus level. In the early stage (1–21 days), the flora in each group mainly comprised *UCG-014*, *Oscillospiraceae*, *Ruminococcaceae*, and *Subdoligranulum* (Figure 11A). In the late stage (22–42 days), the flora in each group mainly comprised *Bacteroidales*, *Clostridia_UCG-014*, *Clostridia_vadinBB60*, *Blautia*, *Barnesiella*, and *Bacteroides* (Figure 11B).

Figure 12 shows the PCA chart of each group at the species level. In the early stage (1–21 days), compared with the CK group, the intestinal microbiota of the PD, LD, MD, and HD groups were similar at the genus level; compared with the PD group, the intestinal microbiota of the LD and HD groups was significantly different (Figure 12A). In the late stage (22–42 days), compared with the CK group, the intestinal microbiota of the PD group, the LD, MD, and HD groups were significantly different at the genus level; compared with the PD group, the microbiota of the LD, MD, and HD groups were similar (Figure 12B).

Figure 13 shows the difference analysis chart for each group at the genus level. In the early stage (1–21 days), compared with the CK group, *Subdoligranulum* showed a significant upward trend in the LD, MD, and HD groups (*p* < 0.05), whereas it decreased significantly in the PD group (*p* < 0.05). *Anaeroplasma* increased significantly in the PD group. *UCG-010* and *UC5-1-2E3* significantly decreased, and *CHKCI002* and *Eubacterium hallii* significantly increased in the LD, MD, and HD groups compared to the CK group (*p* < 0.05) (Figure 13A). In the late stage (22–42 days), *Clostridia_UCG-014* and *NK4A214* showed a significant upward trend compared to the CK group in the LD, MD, and HD groups and in the PD group. The number of *Bacteroidales* increased significantly in the MD and HD groups. Significant decreases in *Erysipelatoclostridium* and *Sellimonas* were found LD, MD, and HD dose groups (Figure 13B).

## 4. Discussion

The main chemical components previously reported for *K. coccinea* are lignans, triterpenes, and flavonoid polyphenols, which was also confirmed by the present study. Studies have shown that lignans from *K. coccinea* root have strong antioxidant function, protective effect on liver injury, and anti-inflammatory effects [19,20]. Triterpenes and flavonoid polyphenols in *K. coccinea* have lipid-lowering, anti-oxidation, anti-tumor effects and inhibitory effects on key enzymes [21,22]. Studies have shown that polysaccharides from *K. coccinea* fruit significantly increase the activities of superoxide dismutase, glutathione peroxidase, and catalase, and decrease the content of malondialdehyde in mice [23]. To date, there have been many reports on the effects of Chinese medicinal materials and plants as feed on broiler growth performance [24,25,26]; however, no previous reports have focused on the effects of *K. coccinea* on the growth performance of broilers. In the present study, we found that in the 1–21 days, each sample does groups and aureomycin group did not show the effect of promoting growth, in the 22–42 days, different does groups of *K. coccinea* fruit extract could improve the growth performance of broilers, and the 200 mg/kg supplementation was the best, having a better effect than that of 300 mg/kg aureomycin, which may have been caused by the adaptation of broilers to the drugs and antibiotics to the efficacy of the drugs and antimicrobials or to the intestinal weakness of broilers at a young age. Studies have shown that adding *Averrhoa bilimbi* L. fruit filtrate, wheat bran, and Saccharomyces cerevisiae to feed of broilers can improve the live body weight and feed conversion ratio [27]. Greene find phytogenic feed additive “comfort” supplementation downregulated the hypothalamic expression of HSP70, reduced core body temperature, increased feed and water intake, and improved body weight in broilers [28]. The content of *K. coccinea* fruit extract in this study, lignans contributed the highest content (28.07%), followed by flavonoids (9.77%), polysaccharide (6.70%), and triterpenoids (5.20%). Therefore, it is speculated that the improvement of *K. coccinea* fruit extract on various aspects of broilers is the result of the interaction of these compounds.

The studies shows that slaughter performance can directly reflect the percentage of the mass of different tissue parts in the total mass and the difference of the deposition amount of nutrients [29] and high abdominal fat in broiler chickens will directly affect the processing of meat products and reduce slaughter rate and consumers’ purchase desire [30]. In this study, we found that PD, LD, MD, and HD groups could increase the dressing percentage, semi-eviscerated percentage, and fully eviscerated percentage and decrease the abdominal fat percentage of broilers. The results indicated that *K. coccinea* fruit extract improved the performance, reduced the fat content of broilers and increased economic benefits.

Studies have shown that adding curcumin and yucca to feed of broilers can improve meat quality by change in the fatty acid profile of meat [31]. Adding fermenting the plant fraction of a solid complete feed (FPFF) to feed of broilers can increase the monounsaturated fatty acid concentration, reduce the cholesterol content and improve the meat quality [32]. In this study, we found that *K. coccinea* fruit extract had no significant effect (*p* > 0.05) on color, drip loss, shear force, and pressure loss of white feathered broilers, but the effects on fatty acid and cholesterol content in broilers require further research.

Studies have shown that increasing the weight of immune organs can enhance immunity, while decreasing their weight can cause immune suppression. The bursal, spleen, and thymus indices are important for evaluating the immune status of chickens [33]. The thymus and bursa of Fabricius are central lymphoid organs of poultry, which are essential to the ontogenetic development of adaptive immunity [34]. The thymus is the organ where T-cells develop, which provide the basis for cellular immunity [35]. In the present study, *K. coccinea* fruit extract could promote the development of immune organs to different extents during the entire rearing period, which indicated that the *K. coccinea* fruit extract may enhance the immunity of broilers.

Serum immunoglobulin is an important indicator of animal immune status, serum immune indices mainly include IgA, IgM, and IgG, which reflect the immunity level of the animal body and indirectly reflect the ability of the body to resist pathogenic microorganisms [36]. C3 is the most abundant complement component in the serum and plays an important role in both classical and bypass complement activation pathways, and C4 is a β-globulin that plays an important role in complement activation, viral neutralization, and phagocytosis [37]. Studies have shown that adding *Summer Savory* (*Satureja hortensis* L.) *Extract* to feed of broilers can improve the levels of IgG and IgM in blood [38]. Liu found that supplementation of 80 mg/kg of natural capsaicin extract in diets could improve the levels of IgG, IgA and IgM in blood [39]. In this study, we found that *K. coccinea* fruit extract and positive drug had different effects on different growth stages of broilers. In the early stage, the levels of IgM and C4 were increased in each sample dose group, while the levels of IgG, IgA and C3 were decreased. In the late stage, most indexes of each dose group were lower or significantly lower than the CK group. In addition, all indexes of PD group were similar or improved to CK group at early growth stage, and all indexes of PD group were better or similar to CK group at late growth stage except IgG. The reason may be the different chemical composition of different additives and their different action methods and pathways and the specific reasons need further study. At the same time, the *K. coccinea* fruit extract can promote the development of immune organs without increasing the level of immune factors, which also needs to be further studied.

Studies have shown that the development of avian immune system is mainly represented by the balanced regulation of Th1/Th2 immune response. Th1 cells mainly divided into IL-2, IFN-y, and TNF-α, which can activate macrophages, mediate cellular immune responses and downregulate immunoglobulin A secretion [40]. After incubation, Th1 immune response is constantly enhanced through stimulation of external antigens, promoting the gradual maturation of immune function [41]. In the present study, we found that *K. coccinea* fruit extracts and positive drugs had different effects on different growth stages of broilers. In the early stage, the expression levels of IL-2 and IFN-γ in each dose group were significantly or extremely significantly decreased, and the expression levels of IL-2 and IFN-γ in the PD group were higher than those in the CK group. However, in the later stage, the expression levels of IL-2 and IFN-γ in the MD and HD group were higher than those in the CK and PD group, and PD group was higher than CK group. The results showed that the expression levels of IL-2 and IFN-γ in the *K. coccinea* fruit extract groups first decreased and then increased, which may have been caused by the adaptability of chicks to the drug, and as the growth time prolonged, Th1 immune response continued to increase, and the immune function gradually matured and was higher than CK and PD groups.

Antioxidants play an important role in maintaining intestinal barrier integrity and preventing bacterial infection [42]. The antioxidant capacity can be expressed by T-AOC, GSH-Px, SOD and MDA etc. in serum [43]. Such as adding *Allium hookeri* to feed of broilers can improve the levels of SOD and CAT, reduce the levels of MDA in blood [44]. Studies have shown that add *Eucommia ulmoides* leaf extract can improve the levels of SOD and reduce the levels of MDA [45]. In the present study, the *K. coccinea* fruit extract of all dose groups can improve the antioxidant function, especially the MD group, the four indexes are better than CK and PD groups.

The digestion and absorption of dietary nutrients occur mainly in the small intestine [46]. The improved feed efficiency in poultry may be partly due to increased nutrient absorption capacity resulting from enhanced intestinal morphology [47]. Previous studies have shown that the addition of *K. alvarezii* (*Kappaphycus alvarezii*) extract to the diet can increase the ratio of velvet to crypto, thus promoting the absorption and digestion of broilers [48]. The addition of a mixture of plant extracts from 500 to 100 mg/kg to the diet increased the chorioto-crypto-ratio of the duodenum, jejunum, and ileum [49]. In this study, in the early stage, *K. coccinea* extract increased the villus height of the jejunum in all dose groups, and reduced the crypt depth in the MD and HD group. Moreover, the ratio of velvet to crypt in the MD and HD group was greater than that in the CK and PD group. In the late stage, the villus height of the jejunum was increased in all dose groups, the crypt depth of the jejunum was decreased in the LD and MD group, and the V/C of all dose groups was greater than that of the CK and PD group. These results indicate that the extract of *K. coccinea* fruit improved the intestinal magnification coefficient, digestibility, and absorption efficiency.

Studies have shown that the abundance of *Subdoligranulum* is positively correlated with microbial richness and HDL-cholesterol levels, and negatively correlated with fat mass, adipocyte diameter, insulin resistance, leptin, insulin, CRP, and IL6 levels in people and animals [50]. Researchers have found that in rabbits, *Subdoligranulum* are useful in the treatment of necrotizing enterocolitis by affecting the production of bacteriophages and butyrate, respectively [51]. *CHKCI002* positively correlated with monococcal and butyric acid production. *Eubacterium hallii* metabolize dietary fiber as a major short-chain fatty acid producer in broiler, providing an energy source for intestinal cells and achieving anti-inflammatory effects in the intestine, and *Eubacterium hallii* can interact with *Bifidobacterium*, leading to the formation of acetate, butyrate, propionate, and formate [52,53,54]. *Clostridia_UCG-014* is a probiotic related to the tryptophan metabolism, positively correlated with fasting blood glucose and inflammation, and is beneficial for the treatment of glucose and lipid metabolism disorder [55,56]. *NK4A214* have a positive correlation in bile acid levels, vitamin A absorption from the gut and was positively related with concentration of sIgA in broiler [57,58]. *Bacteroidales* could promote the existence of IEL in the colon and then produce interleukin-6 (IL-6) in a MyD88 (bone marrow differentiation primary response 88)-dependent manner, thus providing homeostasis for epithelial barrier function [59]. *Erysipelatoclostridium* is pathogenic and can cause erysipeloid, arthritis, and animal erysipelas. In this study, we found that *K. coccinea* fruit extract mainly increased *Subdoligranulum, CHKCI002, Eubacterium hallii, Clostridia_UCG-014, NK4A214* and *Bacteroidales* bacterial flora and decreased *Erysipelatoclostridium* at genus level. In conclusion, *K. coccinea* fruit extract increased the relative abundance of probiotics and beneficial bacteria and reduced the relative abundance of harmful bacteria in the gut of broilers.

## 5. Conclusions

*K. coccinea* fruit extract promotes the growth of broilers; reduces the feed to meat ratio; increases the slaughter performance of broilers, the immune organ index, the relative expression of cytokines, and the body antioxidant capacity; improves jejunal morphology and flora; promotes body health; and has potential as an animal feed additive. Supplementation with 200 mg/kg *K. coccinea* fruit extract had the best comprehensive effect, which was better than that of the 300 mg/kg aureomycin treatment.

## Figures and Tables

**Figure 1 animals-13-00093-f001:**
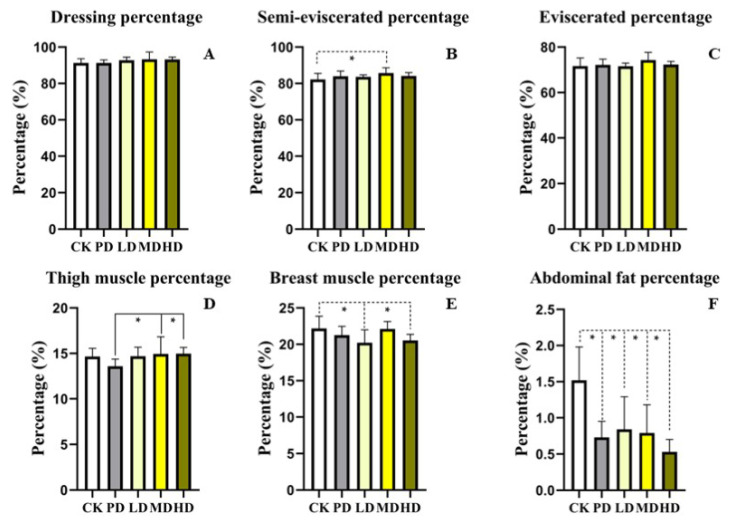
Effects of the fruit extract of *K. coccinea* on slaughter performance of white-feather broilers (**A**) Dressing percentage; (**B**) Semi-eviscerated percentage; (**C**) Eviscerated percentage; (**D**) Thigh muscle percentage; (**E**) Breast muscle percentage; (**F**) Abdominal fat percentage. Note: Dotted line: compare with CK group; Solid line: compare with PD group; * indicate significant differences between groups according to the Tukey test; Appendix A.

**Figure 2 animals-13-00093-f002:**
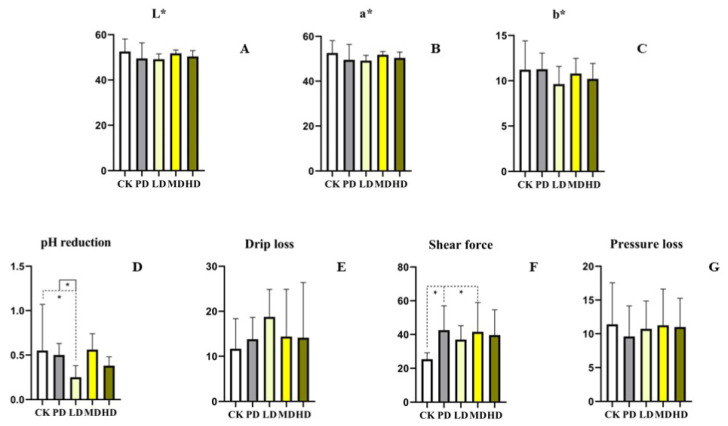
Effects of the fruit extract of *K. coccinea* on leg muscle of white-feather broilers (**A**) L*; (**B**) a*; (**C**) b*; (**D**) pH reduction of 24 h; (**E**) drip loss; (**F**) shear force; (**G**) pressure loss; Note: Dotted line: compare with CK group; Solid line: compare with PD group; * indicate significant differences between groups according to the Tukey test; Appendix A.

**Figure 3 animals-13-00093-f003:**
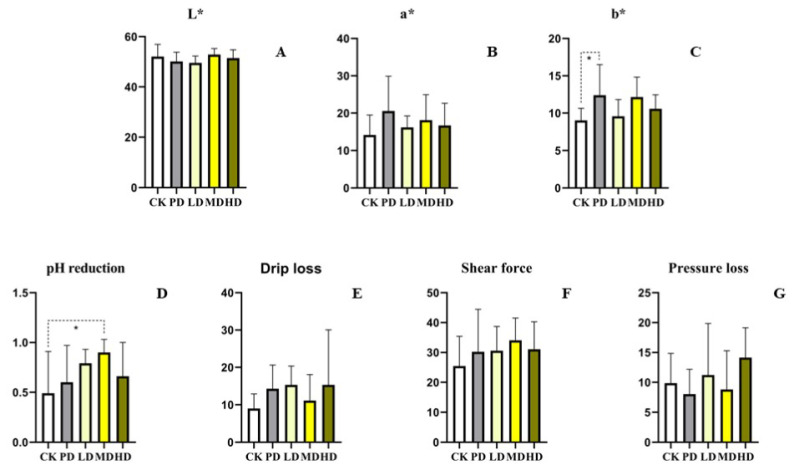
Effects of the fruit extract of *K. coccinea* on chest muscle of white-feather broilers (**A**) L*; (**B**) a*; (**C**) b*; (**D**) pH reduction of 24 h; (**E**) drip loss; (**F**) shear force; (**G**) pressure loss; Note: Dotted line: compare with CK group; Solid line: compare with PD group; * indicate significant differences between groups according to the Tukey test; Appendix A.

**Figure 4 animals-13-00093-f004:**
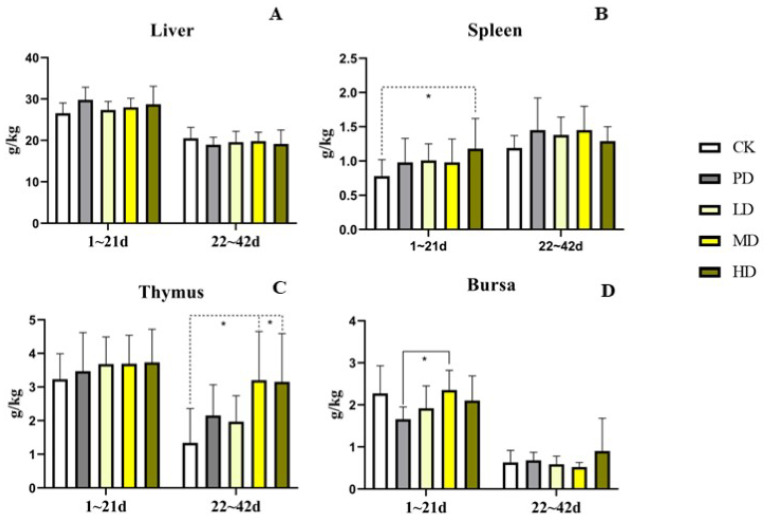
Effects of the fruit extract of *K. coccinea* on immune organ index of white-feather broilers (**A**) liver organ index; (**B**) spleen organ index; (**C**) thymus organ index; (**D**) bursa organ index; Note: Dotted line: compare with CK group; Solid line: compare with PD group; * indicate significant differences between groups according to the Tukey test; Appendix A.

**Figure 5 animals-13-00093-f005:**
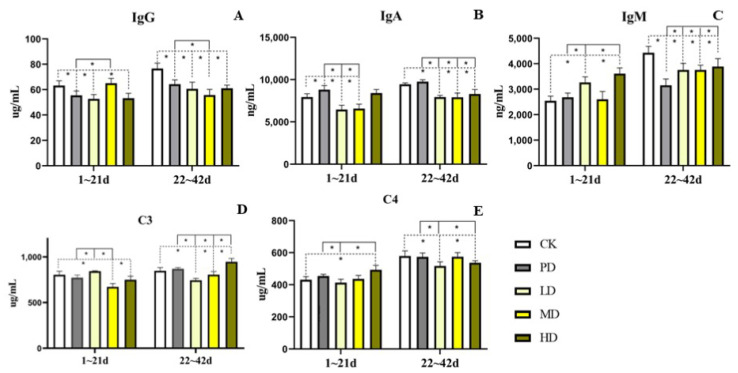
Effects of the fruit extract of *K. coccinea* on immunoglobulin levels in white-feather broilers (**A**) IgG (Immunoglobulin G); (**B**) IgA (Immunoglobulin A); (**C**) IgM (Immunoglobulin M); (**D**) C3 (Complement 3); (**E**) C4 (Complement 4); Note: Dotted line: compare with CK group; Solid line: compare with PD group; * indicate significant differences between groups according to the Tukey test; Appendix A.

**Figure 6 animals-13-00093-f006:**
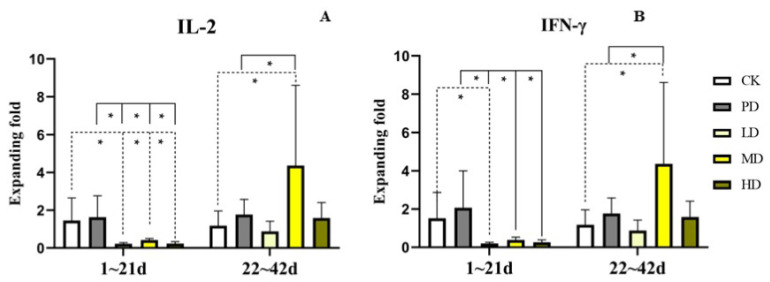
Effects of the fruit extract of *K. coccinea* on mRNA expression levels of cytokines in serum of white-feather broilers (**A**) the expression of IL-2 (interleukin-2); (**B**) the expression of IFN-γ (Interferon alpha-γ); Note: Dotted line: compare with CK group; Solid line: compare with PD group; * indicate significant differences between groups according to the Tukey test; Appendix A.

**Figure 7 animals-13-00093-f007:**
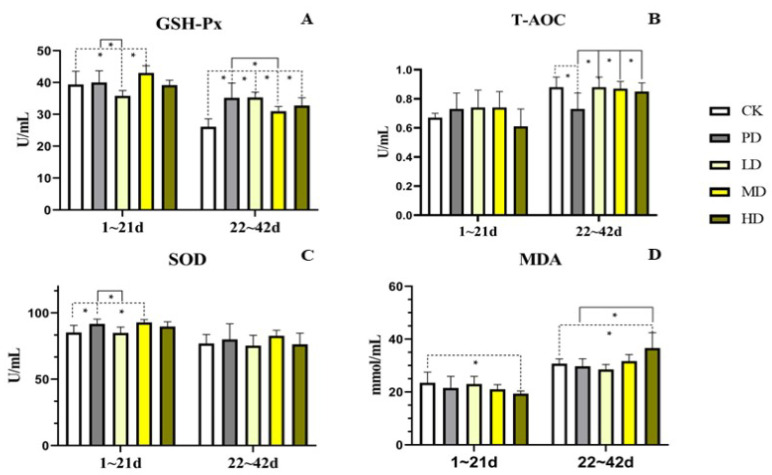
Effects of the fruit extract of *K. coccinea* on antioxidant function of white-feather broilers (**A**) Glutathione peroxidase (GSH-Px); (**B**) Total antioxidant capacity (T-AOC); (**C**) Superoxide dismutase (SOD); (**D**) malonaldehyde (MDA); Note: Dotted line: compare with CK group; Solid line: compare with PD group; * indicate significant differences between groups according to the Tukey test; Appendix A.

**Figure 8 animals-13-00093-f008:**
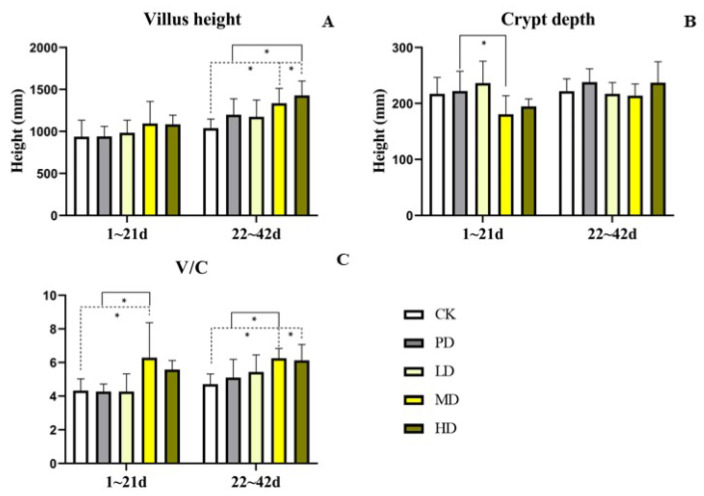
Effects of the fruit extract of *K. coccinea* on antioxidant function of white-feather broilers (**A**) villus height; (**B**) crypt depth; (**C**) The ratio of villus height to crypt depth; Note: Dotted line: compare with CK group; Solid line: compare with PD group; * indicate significant differences between groups according to the Tukey test; Appendix A.

**Figure 9 animals-13-00093-f009:**
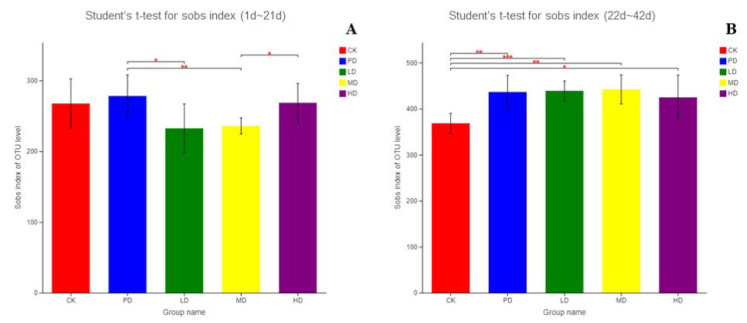
Difference analysis of Alpha diversity between groups (**A**) the early stage (1~21d), (**B**) the late stage (22~42d). Note: * indicate significant differences between groups according to the Tukey test. (*: *p* < 0.05; **: *p* < 0.01; ***: *p* < 0.001).

**Figure 10 animals-13-00093-f010:**
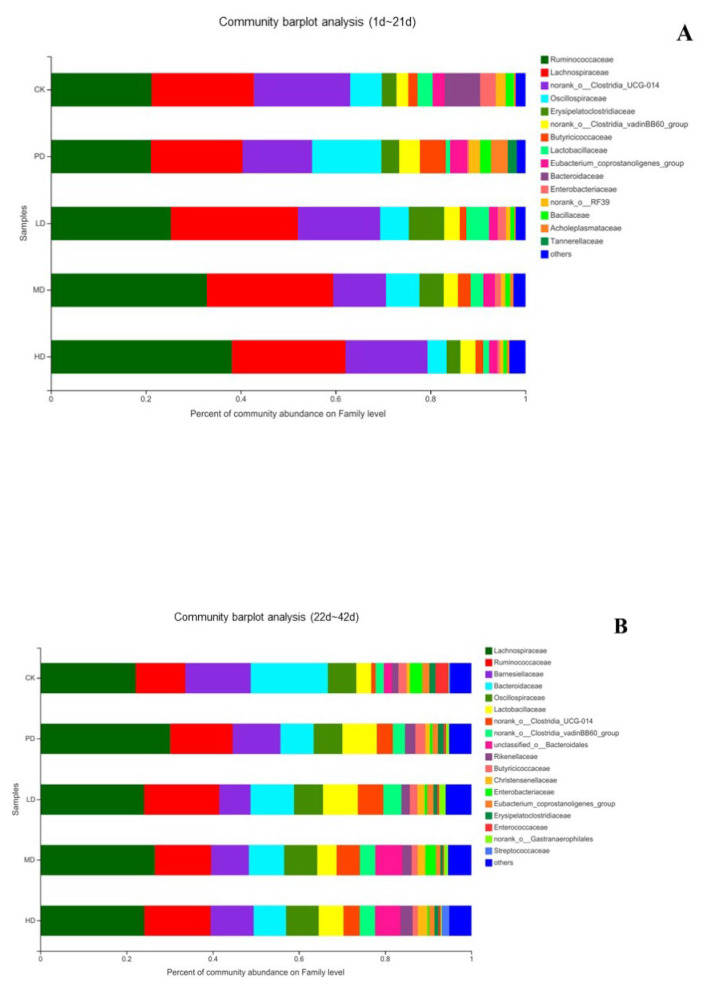
The composition of each group at the family level (**A**) the early stage (1~21 d), (**B**) the late stage (22~42 d).

**Figure 11 animals-13-00093-f011:**
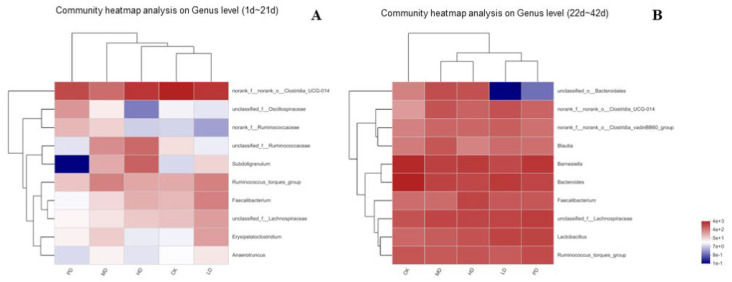
Heat map of the composition of each microbiota at the genus level (**A**) the early stage (1~21 d), (**B**) the late stage (22~42 d).

**Figure 12 animals-13-00093-f012:**
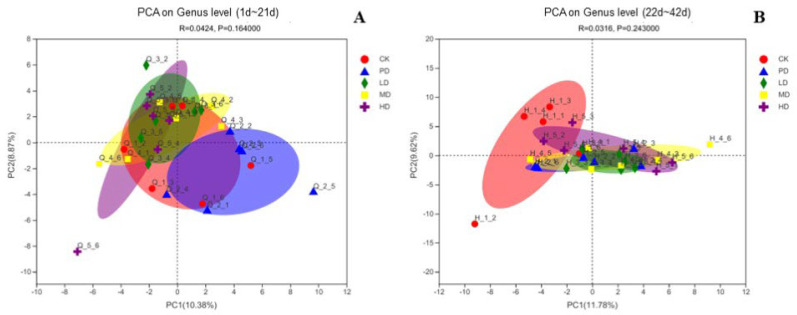
PCA analysis of each group at the genus level (**A**) the early stage (1~21 d), (**B**) the late stage (22~42 d).

**Figure 13 animals-13-00093-f013:**
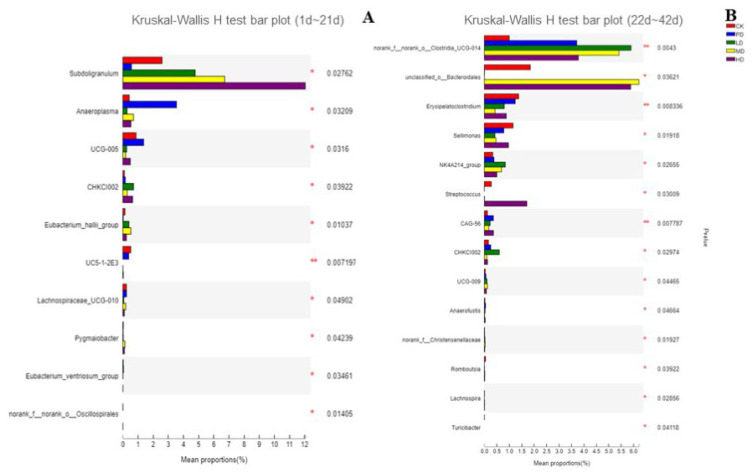
PCA analysis of each group at the genus level (**A**) the early stage (1~21 d), (**B**) the late stage (22~42 d) (*: *p* < 0.05; **: *p* < 0.01).

**Table 1 animals-13-00093-t001:** Experimental basal diet composition and nutrient level.

Ingredients (%)	1~21 d (Early Growth Stage)	22~42 d (Late Growth Stage)
Corn	55.21	62.21
Soybean meal	36.40	29.20
soybean oil	4.70	4.90
Limestone	1.52	1.60
CaHPO4	1.00	1.00
L-Lysine	0.35	0.30
Methionine	0.16	0.13
L-Threonine	0.06	0.06
Salt	0.30	0.30
Premix ^1^	0.30	0.30
Nutrient levels ^2^		
Metabolic energy, MJ/kg	12.72	13.02
Curde protein, %	20.65	18.28
Lysine, %	1.27	1.09
Methionine, %	0.47	0.41
Calcium, %	0.90	0.91
Total phosphorus, %	0.54	0.52

^1^ Premix is supplied per kg of diet: vitamin A 12000 IU, vitamin D 32500 IU, vitamin E 20.0 mg, vitamin K3 3.0 mg, vitamin B1 3.0 mg, vitamin B2 8.0 mg, vitamin B6 7.0 mg, vitamin B12 0.03 mg, pantothenic acid 20.0 mg, niacin 50.0 mg, biotin 0.1 mg, folic acid 1.5 mg, Fe 45 mg, Cu 17.5 mg, I 1.5 mg, Zn 105 mg, Mn 124 mg, Se 15 mg; ^2^ Nutrient level is calculated value.

**Table 2 animals-13-00093-t002:** Contents of main chemical constituents of *K. coccinea* fruit extract (n = 3).

Indicators	Standard Curve	R^2^	Contents (%)
general flavones	y = 0.8566x + 0.0497	0.9963	9.77 ± 0.12
triterpene	y = 2.2275x + 0.0082	0.9918	5.20 ± 0.09
lignans	y = 0.7335x + 0.0055	0.9910	28.07 ± 1.54
polysaccharide	y = 3.8662x + 0.0234	0.9974	6.70 ± 0.23

**Table 3 animals-13-00093-t003:** Effects of the fruit extract of *K. coccinea* on growth performance of white-feather broilers (n = 6).

Items	CK	PD	LD	MD	HD
1~21 d					
Initial weight (g)	46	46	46	46	46
Final weight (g)	708.57 ± 34.83 ^a^	620.56 ± 98.10 ^b^	603.7 ± 95.80 ^b^	680.65 ± 22.28 ^ab^	653.89 ± 66.34 ^ab^
ADG (g)	31.81 ± 1.85 ^a^	25.40 ± 5.26 ^b^	25.40 ± 5.71 ^b^	30.06 ± 1.14 ^a^	28.64 ± 3.08 ^ab^
ADFI (g)	42.59 ± 2.57 ^ab^	40.00 ± 8.13 ^b^	44.10 ± 5.46 ^ab^	46.31 ± 1.53 ^a^	43.10 ± 2.81 ^ab^
F/G	1.34 ± 0.05 ^b^	1.53 ± 0.19 ^b^	1.82 ± 0.49 ^a^	1.54 ± 0.36 ^ab^	1.51 ± 0.13 ^b^
22~42 d					
Final weight (g)	2476.67 ± 70.90 ^b^	2607.69 ± 384.75 ^ab^	2510.91 ± 217.27 ^ab^	2739.20 ± 105.95 ^a^	2664.35 ± 81.33 ^ab^
ADG (g)	83.70 ± 3.21 ^b^	93.97 ± 14.32 ^a^	87.64 ± 5.76 ^ab^	97.10 ± 6.28 ^a^	91.87 ± 5.74 ^ab^
ADFI (g)	134.03 ± 11.67 ^ab^	140.92 ± 18.39 ^ab^	125.81 ± 11.53 ^b^	142.03 ± 14.81 ^ab^	149.16 ± 11.62 ^a^
F/G	1.60 ± 0.16 ^ab^	1.51 ± 0.13 ^ab^	1.44 ± 0.19 ^b^	1.47 ± 0.15 ^ab^	1.62 ± 0.10 ^a^
1~42 d					
ADG (g)	57.87 ± 1.69 ^b^	60.99 ± 9.16 ^ab^	58.69 ± 5.17 ^ab^	64.12 ± 2.52 ^a^	62.34 ± 1.94 ^ab^
ADFI (g)	88.31 ± 6.86 ^ab^	90.49 ± 11.14 ^ab^	84.95 ± 6.11 ^b^	94.17 ± 7.23 ^a^	96.13 ± 5.01 ^a^
F/G	1.53 ± 0.10	1.49 ± 0.07	1.46 ± 0.22	1.46 ± 0.09	1.54 ± 0.08

Different letters in the same column indicate significant differences between groups according to the Tukey test; ^a,b^ Means in the same row with different superscript letters indicate differences (*p* < 0.05).

## Data Availability

Raw data are held by the authors and may be available upon request.

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
