# Peer review of "Effects of Kadsura coccinea L. Fruit Extract on Growth Performance, Meat Quality, Immunity, Antioxidant, Intestinal Morphology and Flora of White-Feathered Broilers"

_animals, 2022, doi:10.3390/ani13010093_

Round 1
Reviewer 1 Report
The authors found an interesting and useful work. The study looks well planned and conducted. I suggest only independent changes.
1. A brief summary should be supplemented with more information that will allow the lay reader to understand how relevant the topic of phytobiotics research is, in general, and in particular yours. I think I should add here why Kadsura coccinea fruit extract was chosen for you (briefly).
2. The annotation should be rewritten, because after reading it, a not very clear idea of the content of the article is created. Also, in my opinion, it is necessary to supplement the significant results that you received during the research.
Try to avoid abbreviations in this section as well.
3. In the maintenance section, it is necessary to add references to the literature, in my opinion, what is presented is not enough. In particular, it is better to rewrite the sentence somewhat differently "Antibiotics are a class of secondary metabolites produced by microorganisms (including bacteria, fungi, and actinomycetes), animals, and plants, which have antipathogenic effects and interfere with the development of other living cells" (line 34-36). It is not entirely clear which animals produce antibiotics? Or you should provide a more detailed reference to the scientific literature.
4. It is not entirely clear on which breed or cross the birds conducted the research? They were probably broiler chickens.
5. In section 2.1, references should be given to the literature on the basis of which literature you made extracts.
6. Section 2.2 should provide a more detailed analysis.
7. Sections must be completed according to the requirements of the journal.
8. It is not entirely clear from the material and research methods on which analyzer you determined Antioxidant indicators and immune function (line 146)
9. The results of Chapter 3.1 should be expanded. Or present it in the form of a table. Sections do not consist of a single sentence.
10. It is better to present Figure 1 in the form of a table.
11. Figure 2 should be presented in a different form. In the form in which it is presented is unpresentable.
12. Figure 3 it is not entirely clear where "A" goes and where "B" goes. A similar remark with paragraph 11.
13. It is better to make Figure 10 larger. There is no visible decoding of microorganisms.
14. Please provide the number and date of the Bioethics committee protocol at the end of the article. And it is necessary to make a similar signature after the inscription of section No. 2.
Reviewer 2 Report
The language should be improved for better understanding of the work. At places it was very difficult to understand the content.
The work is good and very elaborate
Introduction is good and gives a clear indication of the kind of undertaken and its importance
Material methods have been given well but sample size should be discussed properly for each experiment.
Results: clearly presented
Discussion is weak. This section requires thorough improvement for justifying the work or its relevance.
References: fine

Reviewer 3 Report
The aim of the study was to investigate the effects of K. coccinea fruit extract on growth performance, meat quality, slaughter performance, immune function, cytokines, antioxidant capacity, intestinal morphology, and intestinal microbiota of white-feathered broilers. The subject of the manuscript falls within the general scope of the journal. The experiment has been well planed. The results are interesing and important for the poultry producers. In my opinion, the manuscript could be suitable for publication after revision, addressing the following comments:
Throughout the manuscript
Please carefully check using a space in the text. The Latin names of the species and other terms in this language should be italicized.
Simple Summary
Line 13: „It can improve … meat quality …”. The presented results did not show this.
Lines 14-15: “This has also met consumers' expectations for the quality of broiler meat.”. Please indicate the results that confirm this.
Keywords
„antibiotics” - I suggest to remove.
Introduction
Lines 34-36: Please insert the references.
Lines 38-39: Please add more information about use antibiotics in animal husbandry (numerical data).
Lines 44-45: Please indicate the countries have legislated the restriction or even prohibition of the addition of antibiotics to feed.
Lines 45-49: Please insert the references.
Lines 49-51: The studies on the use various herbs as a natural antibiotics in animal husbandry are being conducted around the world.
Lines 52-55: Please insert the references.
Line 57: „In our previous study …” - please insert the references.
Lines 62-63: I suggest to remove. This was not the aim of the research.
Material and methods
Lines 82-83: Please describe more details of the method determine components of K. coccinea fruit extracts.
Line 85: „…broilers with similar body weights …”. The birds were weighted?
Line 112: Please separate the subsections.
Line 127: Please replace „animals” with „broilers” or „birds”.
Line 128: „… the hair was sched …” Please explain.
Lines 131-132: All detailed methods used should be described in the present manuscript, because they are described in Chinese ” (NY/T 823-2004, China).
Lines 134-140: Please describe the methods in more detail and insert the relevant references.
Lines 142-147: Please describe the methods in more detail and insert the relevant references.
Line 158: Please explain the abbreviation used the first time - Ct (cycle threshold).
Results
When you put significance value of difference, please not use word „significantly”.
Figures are hard to read. Abbreviations in figures legends should be explained.
Please see Statistical Analysis section and the titles figures 1-8. Please explain: significances were set at P<0.05 level or maybe P<0.05 and P<0.01.
Lines 216-220: Methods of measure of the shear force and pH24 value have not been described in the Materials and Methods section.
Discussion
The authors should try to more explain the obtained results. Moreover, the results of presented in this manuscript have not been compared with the results of other studies enough. Authors should avoid repeating research results in this section.
Lines 384-388: I suggest to remove this part. The Authors should more relate to the discussion of the results in this section, not to presentation of general information.
Lines 396-400: Please see comments to lines 384-388.
Lines 414-421: Please see comments to lines 384-388.
Lines 497-515: I suggest moving this paragraph to the beginning of the chapter.
Reviewer 4 Report
please check the attached file

Round 2
Reviewer 4 Report
Good paper
Author Response
Dear Dr.,
Thank you for your decision letter on 12/21/2022 and the reviewers’ comments concerning our manuscript entitled as “Effects of Kadsura coccinea L. fruit extract on growth performance, meat quality, immunity, antioxidant, intestinal morphology and flora of white-feathered broilers”. We have checked the English language and style. We uploaded the files of the revised manuscript. We believe that our revised manuscript is significantly improved and acceptable for publication in Animals.
We would like to thank you for giving us an opportunity to resubmit this revised copy of the manuscript and we really appreciate your time and consideration. We look forward to a positive decision.
Best regards,
Tianlu Zhang
On behalf of all authors
Email: ztl15581233625@163.com